# Relationship between infant gastrointestinal microorganisms and maternal microbiome within 6 months of delivery

Menglu Li,[1] Yuling Xue,[2] Han Lu,[1] Jinping Bai,[1] Liru Cui,[1] Yibing Ning,[2] Qingbin Yuan,[2] Xianxian Jia,[3] Shijie Wang[1,2]

**ABSTRACT** To investigate the association between the microbiota in mothers and gut microbiota in infants from 0 to 6 months, the microbiotas in infant feces, maternal feces, and breast milk were determined by 16S rRNA gene sequencing. The contribution of each maternal microbiome to the infant was assessed using fast expectation-maximization for microbial source tracking calculations. The levels of short-chain fatty acids (SCFAs) and secretory immunoglobulin A (sIgA) in the feces of infants were also determined using gas chromatography and IDK-sIgA ELISA to gain a more comprehensive understanding of the infant gut microbiome. The results of this study showed that in addition to Firmicutes (E1) and *Bifidobacterium* (E2), the dominant microorganisms of the intestinal microbiota of infants aged 0–6 months include Proteobacteria, which is different from previous findings. Acetic acid, the most abundant SCFA in the infant gut, was positively correlated with *Megasphaera* ($P < 0.01$), whereas sIgA was positively correlated with *Bacteroides* ($P < 0.05$) and negatively correlated with *Klebsiella* and *Clostridium_XVIII* ($P < 0.05$). The maternal gut microbiota contributed more to the infant gut microbiota ($43.58\% \pm 11.13\%$) than the breast milk microbiota, and significant differences were observed in the contribution of the maternal microbiota to the infant gut microbiota based on the delivery mode and feeding practices. In summary, we emphasize the key role of maternal gut health in the establishment and succession of infant gut microbiota.

**IMPORTANCE** This study aims to delineate the microbial connections between mothers and infants, leveraging the fast expectation-maximization for microbial source tracking methodology to quantify the contribution of maternal microbiota to the constitution of the infant's gut microbiome. Concurrently, it examines the correlations between the infant gut microbiota and two distinctive biomolecules, namely short-chain fatty acids (SCFAs) and secretory immunoglobulin A (sIgA). The findings indicate that the maternal gut microbiota exerts a greater influence on the infant's gut microbial composition than does the microbiota present in breast milk. Infants born via vaginal delivery and receiving mixed feeding display gut microbiota profiles more similar to their mothers'. Notably, the SCFA acetate displays positive associations with beneficial bacteria and inverse relationships with potentially harmful ones within the infant's gut. Meanwhile, sIgA positively correlates with Bacteroides species and negatively with potentially pathogenic bacteria. By delving into the transmission dynamics of maternal-infant microbiota, exploring the impacts of metabolic byproducts within the infant's gut, and scrutinizing how contextual factors such as birthing method and feeding practices affect the correlation between maternal and infant microbiota, this research endeavors to establish practical strategies for optimizing early-life gut health management in infants. Such insights promise to inform targeted interventions that foster healthier microbial development during the critical first 6 months of life.

**KEYWORDS** 0-6-month infant, gut microbiome, maternal, human milk, feces

Address correspondence to Shijie Wang, wangshijie@hebust.edu.cn.

Menglu Li and Yuling Xue contributed equally to this article. Author order was determined on the basis of contribution.

The authors declare no conflict of interest.

See the funding table on p. 14.

The intestinal microbiota is a set of microbial communities in the digestive tract that plays a significant physiological role in the health of the whole body (1–3). Infants have dynamic and highly personalized microbial traits compared to adults, and infancy is a crucial time for the establishment and development of the gut microbiota (4). After birth, the intestinal microbiota develops over a protracted period of time from a sparse colony of bacteria to an abundance of various species, and it eventually stabilizes and approaches that of adults while maintaining a dynamic and stable balance (5, 6). The origin of the infant gut microbiota has been debated for many years. Although the origin and mechanisms of colonization are unknown, studies have implied that the mother is the baby's initial source of microorganisms (7, 8). Moreover, studies have suggested that the environment and the mother's birth canal and excretions are significant sources of microbes for infants (9).

The maternal diet, delivery mode, feeding mode, and antibiotics use can affect the development of the infant's microbiome from birth onward (10–12). Although numerous studies have reported the variables affecting infant gut microbiota, source analyses of infant gut microbiota and the relative contributions of each source have not been explained in detail. Newborns are exposed to microorganisms in the mother's womb and acquire them through vaginal delivery or cesarean section and breastfeeding. A significant association is observed between microorganisms carried by babies at birth and those carried by their mothers (13, 14). Although conclusive data have not been obtained on the vertical transfer of the composition of gut microbiota in early infancy, the first-colonizing microbes that infants acquire after birth are likely to colonize their gut for many years (15).

Short-chain fatty acids (SCFAs) and secretory immunoglobulin A (sIgA) are two materials that have attracted particular interest in recent years in relation to the gut microbiota (16–21). SCFAs are primarily produced in the human body by the intestinal microbiota when they break down carbohydrates and other substances in food (22). The major SCFAs in the adult body are acetic acid (AA), propionic acid (PA), and butyric acid (BA), which account for more than 90% of the total acids and have a ratio of 3:1:1 (16, 17). Nonetheless, the content of SCFAs within the intestinal milieu exhibits significant inter-individual variation among infants, differing notably from that observed in adult subjects, thereby highlighting an age-dependent disparity in the gut's metabolic profile (23). Numerous factors affect the quantity of SCFAs, and the amount of SCFAs in the infant's gut is directly correlated with the baby's full-term status, delivery method, and feeding schedule (24, 25). The amount of SCFAs in host stools has been measured to diagnose intestinal diseases, such as celiac disease, inflammatory bowel disease, colorectal cancer, and neonatal necrotizing small bowel colitis (25). sIgA levels are crucial for neonatal health and play a critical role in the innate immune system by protecting against pathogens. Most of the body's sIgA-transporting plasma cells are found in the intestinal mucosa, and these cells release many grams of sIgA into the intestine daily (26). Although the precise source of sIgA in the infant intestine remains unknown, such sources can be divided into two main categories: endogenous and maternal. Infants do not produce endogenous IgA in the first month of life and thus can only obtain IgA from breast milk at this time. Therefore, the body's immune system and early newborn gut microbiota development depend on sIgA (27).

Studies comparing the colonization of the infant's gut by microorganisms from different maternal sites have been published. For example, microorganisms found on the mother's skin and vagina colonize the infant gut only briefly, whereas microorganisms from the mother's gut colonize the infant's gut more permanently (28). Although some potential links between maternal microbiota and infant gut microbiota have been reported in the literature, the specific values for potential sources of infant gut microbiota have not been characterized during the first 6 months of life when breast milk is the only source of nutritional intake. Therefore, this study focuses on the effect of both the breast milk microbiota and maternal gut microbiota on the infant gut microbiota after 6 months of life and analyzes the infant gut microbiota during this time period.

## RESULTS

### Participant profile

All participants in this study completed questionnaires at the time of sample collection, which included basic physical information and information related to their reproductive history (Table 1). The mean age of the women was 29 ± 4 years, the mean height was 160.41 ± 5.01 cm, the weight before pregnancy was 68.39 ± 12.28 kg, and the weight after delivery was 61.22 ± 10.41 kg. Most babies were born via vaginal delivery (64.16%), and the feeding method was breastfeeding (85.37%). Most samples were collected from both mothers and infants at the same time, resulting in 41 infant, 38 mother, and 41 breast milk samples.

### Gut microbiota analysis of infants aged 0–6 months and its correlation with special substances in feces

First, the gut microorganisms of infants aged 0–6 months were classified into enterotypes. The classification results are shown in Fig. 1A. An evolution of our enterotypes of the intestinal microbiota in infants was observed from birth to 3 years of age, and the microbiota that play a dominant role included Firmicutes (E1), *Bifidobacterium* (E2), Bacteroidetes (E3), and *Prevotella* (E4) (29). The intestinal microbiota of 0–6-month-old infants contained the same E1 and E2. Notably, the presence of Proteobacteria was discerned, which deviates from findings reported in prior investigations. This discordance may be attributed to the fact that our study specifically focused on infants within the narrow age bracket of 0–6 months. Proteobacteria are dominant during the nascent stages of life, only to be progressively supplanted by other microbial constituents as the host matures and develops.

To further understand the specific composition of the gut microbiota of infants from 0–6 months, the infant samples were analyzed for the composition of core microorganisms, which were defined as genera with an average relative abundance of ≥1% that were present in at least 95% of the samples (Table 2) (30, 31). Based on the above requirements, 11 core microorganisms belonging to four major phyla were identified in the infant samples: Firmicutes, Actinobacteria, Bacteroidetes, and Proteobacteria. To better visualize the composition of infant gut microorganisms, we conducted a cluster analysis of infant gut core microorganisms using the Spearman correlation coefficient and identified four clusters (Fig. 1C). The four major clusters were separated based on their relative abundance and included Enterobacteriaceae and Bifidobacteriaceae

**TABLE 1** Features of the study's 41 lactating participants and their offspring[a]

| Participants/offspring features | Measurement |
| --- | --- |
| Age (yr) | 29 ± 4 |
| Ht (cm) | 160.41 ± 5.01 |
| Prenatal wt (kg) | 68.39 ± 12.28 |
| Postpartum wt (kg) | 61.22 ± 10.41 |
| Postpartum BMI | 23.80 ± 4.04 |
| Gestational wk | 39.27 ± 1.05 |
| Mode of delivery | |
| Vaginal delivery (no., %) | 27 (64.16) |
| Cesarean delivery (no., %) | 14 (36.84) |
| Infant gender | |
| Male (no., %) | 20 (48.78) |
| Female (no., %) | 21 (51.22) |
| Feeding mode | |
| Exclusive BF (no., %) | 35 (85.37) |
| Mixed feeding (no., %) | 6 (14.63) |

[a]BMI, body mass index; BF, breast feeding.

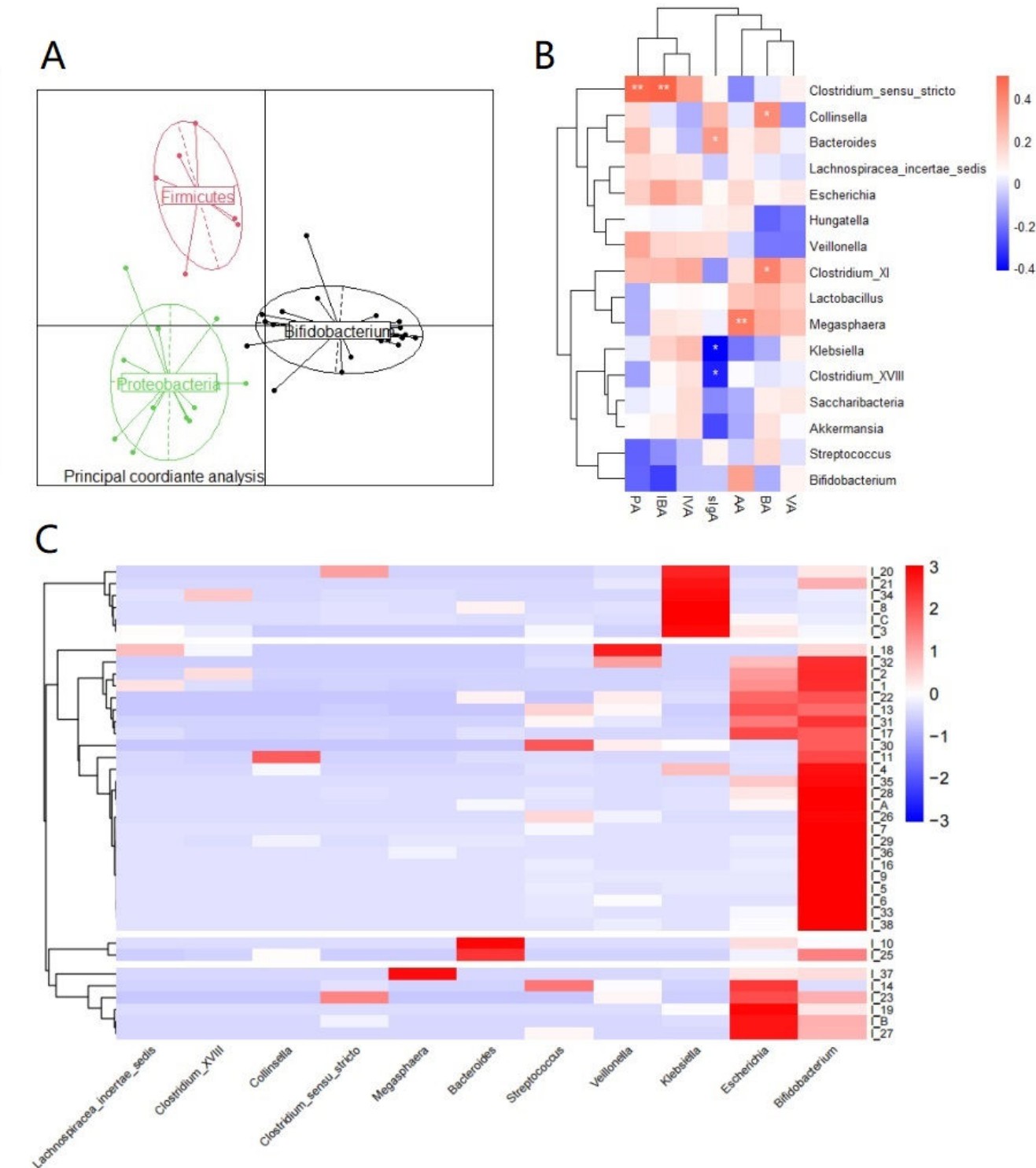

**FIG 1** Classification of gut microbiota in 0–6-month-old infants and correlation with key substances in feces. (A) Classification of gut microbiota in 0–6-month-old infants based on principal component analysis. (B) Correlation heatmap of dominant gut microbiota genera with SCFAs and sIgA in feces. (C) Cluster analysis of core microbiota in infant gut microbiota.

(cluster 1, $n = 6$), Bifidobacteriaceae and Veillonellaceae (cluster 2, $n = 24$), Bacteroidaceae (cluster 3, $n = 2$), and Enterobacteriaceae and Veillonellaceae (cluster 4, $n = 6$).

**TABLE 2** Core microbiota of the intestine of 0–6-month-old infants[a]

| Source | Genus | Coverage (%) | Relative abundance (%) |
|---|---|---|---|
| *Firmicutes-Lachnospiraceae* | *Lachnospiracea_incertae_sedis* | 100.00 | 1.02 ± 3.49 |
| *Firmicutes-Erysipelotrichaceae* | *Clostridium_XVIII* | 95.12 | 1.11 ± 3.85 |
| *Actinobacteria-Coriobacteriaceae* | *Collinsella* | 100.00 | 1.52 ± 6.65 |
| *Firmicutes-Clostridiaceae* | *Clostridium_sensu_stricto* | 100.00 | 1.72 ± 5.92 |
| *Firmicutes-Veillonellaceae* | *Megasphaera* | 97.56 | 1.81 ± 9.96 |
| *Bacteroidetes-Bacteroidaceae* | *Bacteroides* | 100.00 | 3.69 ± 12.53 |
| *Firmicutes-Streptococcaceae* | *Streptococcus* | 100.00 | 3.81 ± 7.64 |
| *Firmicutes-Veillonellaceae* | *Veillonella* | 100.00 | 3.99 ± 8.13 |
| *Proteobacteria-Enterobacteriaceae* | *Klebsiella* | 100.00 | 11.05 ± 22.87 |
| *Proteobacteria-Enterobacteriaceae* | *Escherichia* | 100.00 | 15.54 ± 18.52 |
| *Actinobacteria-Bifidobacteriaceae* | *Bifidobacterium* | 100.00 | 42.78 ± 29.00 |

[a]Source refers to the composition of core OTUs belonging to phylum and family. Coverage represents the coverage of the core genus in the sample.

Due to the unique characteristics of the infant population, we collected infant feces to complete our study because we only intended to employ *in vitro* experimental methods to generate data on the digestive system in the early stages of life. The amount, consistency, and color of the infant feces were described using the Amsterdam infant stool scale (32). The infant feces differed in shape and primarily had the following values: amount = 2, consistency = B, and color = II. Normal adult stools have a pH between 6.9 and 7.2 and correspond to neutral, weakly acidic, or weakly alkaline levels. Because of the abundance of *Lactobacillus* and *Bifidobacterium* in the colon of newborns who have been breastfed for a long period and whose bacteria break down nutrients to produce large amounts of acids, the pH of infant stool is frequently less than 5.5 (Table 3). AA predominated in the early intestinal SCFAs, accounting for 70% of the total acids, with PA, BA, valeric acid (VA), and other branched-chain fatty acids accounting for the remaining 20%. We then assessed the relationship between the dominant genera and SCFAs in the gut microbes using Spearman's coefficient and found that *Megasphaera* showed a highly significant positive correlation with AA ($P < 0.01$), *Clostridium_sensu_stricto* showed a highly significant positive correlation with PA ($P < 0.01$), and *Collinsella* and *Clostridium_XVIII* showed a significant positive correlation with BA ($P < 0.5$). sIgA is a common antibody of the human immune system, and sIgA antibodies offer an immune defense against benign bacteria found in the intestinal flora that would otherwise have negative consequences on the health of the organism. As shown in Fig. 1B, *Bacteroides* displayed a substantial positive association with sIgA content ($P < 0.05$), whereas *Klebsiella* and *Clostridium_XVIII* displayed a significant negative correlation with sIgA content ($P < 0.05$).

**TABLE 3** Characteristics of stool from 0- to 6-month-old infants[a]

| Stool characteristics | Measurement |
|---|---|
| pH | 5.23 ± 0.55 |
| sIgA (mg/mL) | 10.14 ± 9.03 |
| SCFAs (µg/mL) | |
| AA | 1154.05 ± 742.91 |
| PA | 227.93 ± 333.41 |
| BA | 42.84 ± 77.48 |
| VA | 15.85 ± 18.69 |
| IBA | 42.84 ± 77.73 |
| IVA | 43.03 ± 61.84 |
| TA | 1652.62 ± 788.99 |

[a]IBA, isobutyric acid; IVA, isovaleric acid; TA, total acid.

## Infant feces, maternal feces, and breast milk microbiomes

Paired maternal and infant samples were collected, accumulating a total of 120 specimen sets (Fig. 2A). The microbial populations in breast milk (MM), maternal feces (MF), and baby feces (IF) differed, and the four major phyla that accounted for more than 90% of the relative abundance of the intestinal flora were illustrated in Fig. 2B. At the phylum level, the microorganisms in the IF, MF, and the MM all differed noticeably, with the

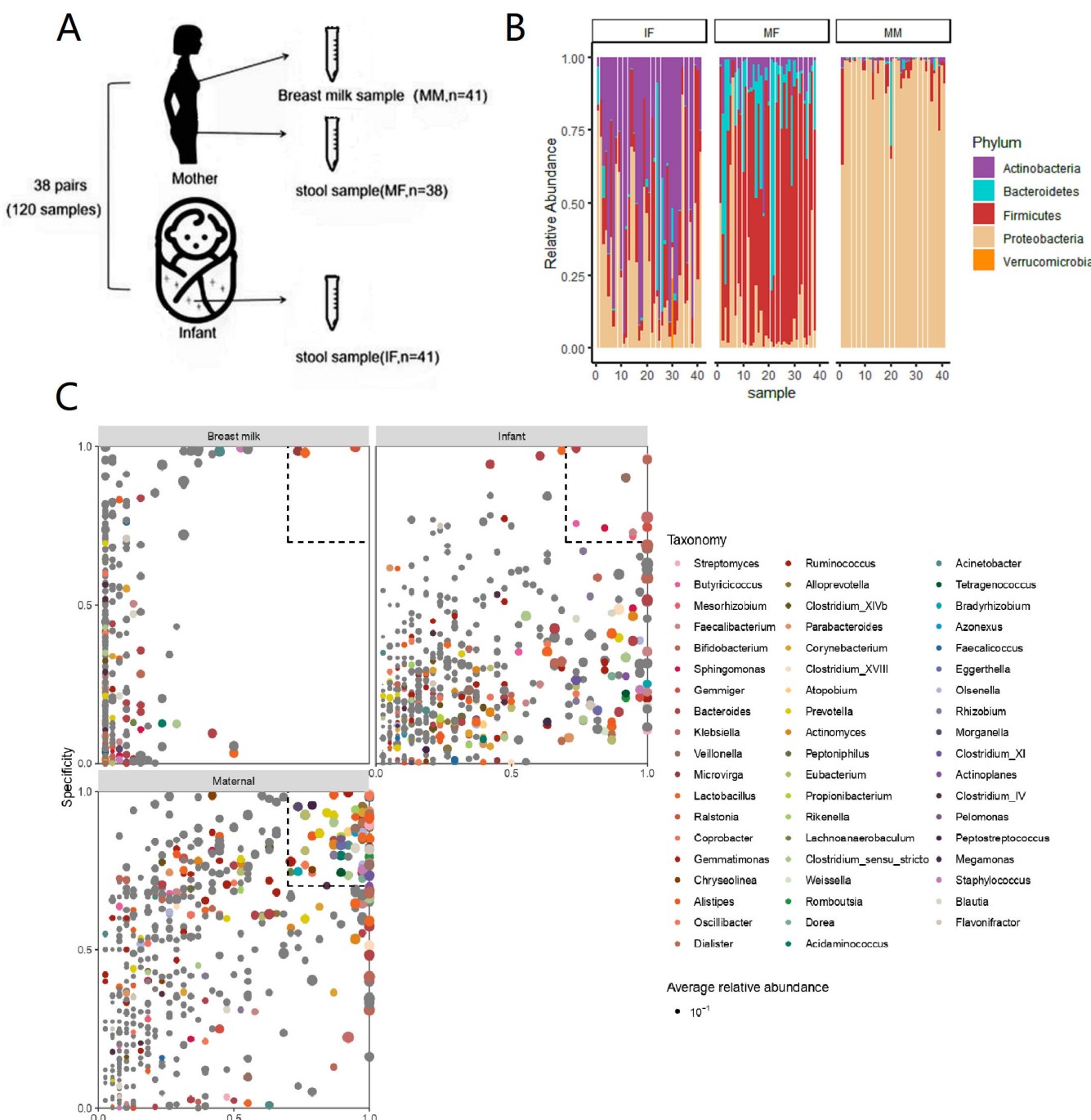

**FIG 2** Thorough analysis of mother–infant pairs at the phylum and genus levels. (A) Methods for collecting samples from mothers and infants. Samples include infant feces (IF), mother's feces (MF), and breast milk (MM). (B) Maternal and infant microbial composition (relative abundance) at the level of phylum. (C) Microbiota (genus level) with specificity and occupancy >0.7 are selected as the targets, and the SPEC-OCCU method is used to represent the main microbiota (unique OTUs) in the maternal and infant samples.

IF predominately composed of Actinobacteria (mean ± SD, relative abundance: 45.22% ± 29.72%, range: 2.45%–94.72%) and Proteobacteria (mean ± SD, relative abundance: 29.45% ± 25.15%, range: 1.34%–87.05%); the MF predominately composed of Firmicutes (mean ± SD, relative abundance: 61.94% ± 24.96%, range: 5.72%–93.79%) and Bacteroidetes (mean ± SD, relative abundance: 17.46% ± 21.35%, range: 0.19%–78.92%); and the MM predominately composed of Proteobacteria (mean ± SD, relative abundance: 91.81% ± 9.70%, range: 58.50%–99.87%). To examine the unique OTUs in IF, MF, and MM, we chose genera with specificity and occupancy >0.7 as the target (Fig. 2C). Specificity-Occupancy (SPEC-OCCU) revealed that the infant's colon contains 10 unique bacterial genera, including *Klebsiella*, *Veillonella*, *Bacteroides*, and *Bifidobacterium*. Forty-eight unique bacterial genera were present in the maternal gut, including *Lactobacillus*, *Faecalibacterium*, and *Parabacteroides*. Only three genera of bacteria were covered by our standards when present in human milk: *Lactobacillus*, *Ralstonia*, and *Microvirga*. Human milk microbes and newborn intestinal microbiota were less varied and complex than those of the mother.

## Maternal and infant sample α- and β-diversity analysis and potential sources of infant intestinal flora

There was substantial variation in α-diversity among the three samples (Fig. 3A and B). We employed the Chao1 and Inverse Simpson indices to assess differences in α-diversity among IF, MF, and MM. The analysis results indicate that the newborn gut microbiota richness was significantly lower than that of the maternal gut microbiota ($P < 0.001$). The Chao1 index was used to indicate species richness in the community. Utilizing the Inverse Simpson index as a proxy for quantifying species diversity within the microbial community, it became evident that the gut microbiota of infants during the early stages of life exhibited a significantly diminished biodiversity compared to that of both the maternal gut microbiota and the microbial inhabitants of breast milk ($P < 0.0001$). This indicates that the low abundance and diversity of the gut microbiota at the beginning of life are consistent with previously reported results. Because the high species proximity of the samples made it easier to identify differences between them, the Bray–Curtis distance was selected for the principal coordinate analysis (PCoA; β-diversity). Infant gut microbes in mother–infant samples were clearly separated from the maternal gut microbiota and breast milk microbes (Fig. 3C). Because of the intimate relationship between mother and infant and the evidence for vertical transmission of microbiota between mother and infant, fast expectation-maximization for microbial source tracking (FEAST) software was used to explore the potential sources of the infant's gut microbes. Using the common amplicon sequences (ASVs) of the three samples, namely IF (sink), MF (source), and MM (source), as the number of categorical units for data reading, we assessed the contributions of maternal gut and breast milk microbiota to the establishment of the infant gut microbiome. The data from the software analysis showed that the mother's gut (43.24% ± 11.16%) was the primary source of gut microbes in newborns at the start of life, with breast milk contributing less (23.01% ± 0.47%; Fig. 3D). Given that an infant has the closest relationship with its mother, who is with the child virtually constantly, it makes sense that the infant's gut microbiota is most similar to that of the mother. Breast milk is the main meal consumed by newborns aged 0–6 months, and it is digested and absorbed by organisms after it is consumed by the infant. During this process, some bacteria may fail to colonize the infant's body.

## Analysis of the contribution of maternal samples to infant gut microbiota

We proceeded with the FEAST analysis of the sequencing results derived from a rigorously matched set of 38 mother–infant pairs. Maternal gut microbiota and breast milk microbes were used as two fixed sources to investigate microbial transmission between maternal and infant samples in more detail, and the differences in the contribution of these two sources to the gut microbes of infants with various delivery and feeding methods were analyzed (Fig. 4). Infants delivered vaginally exhibited a

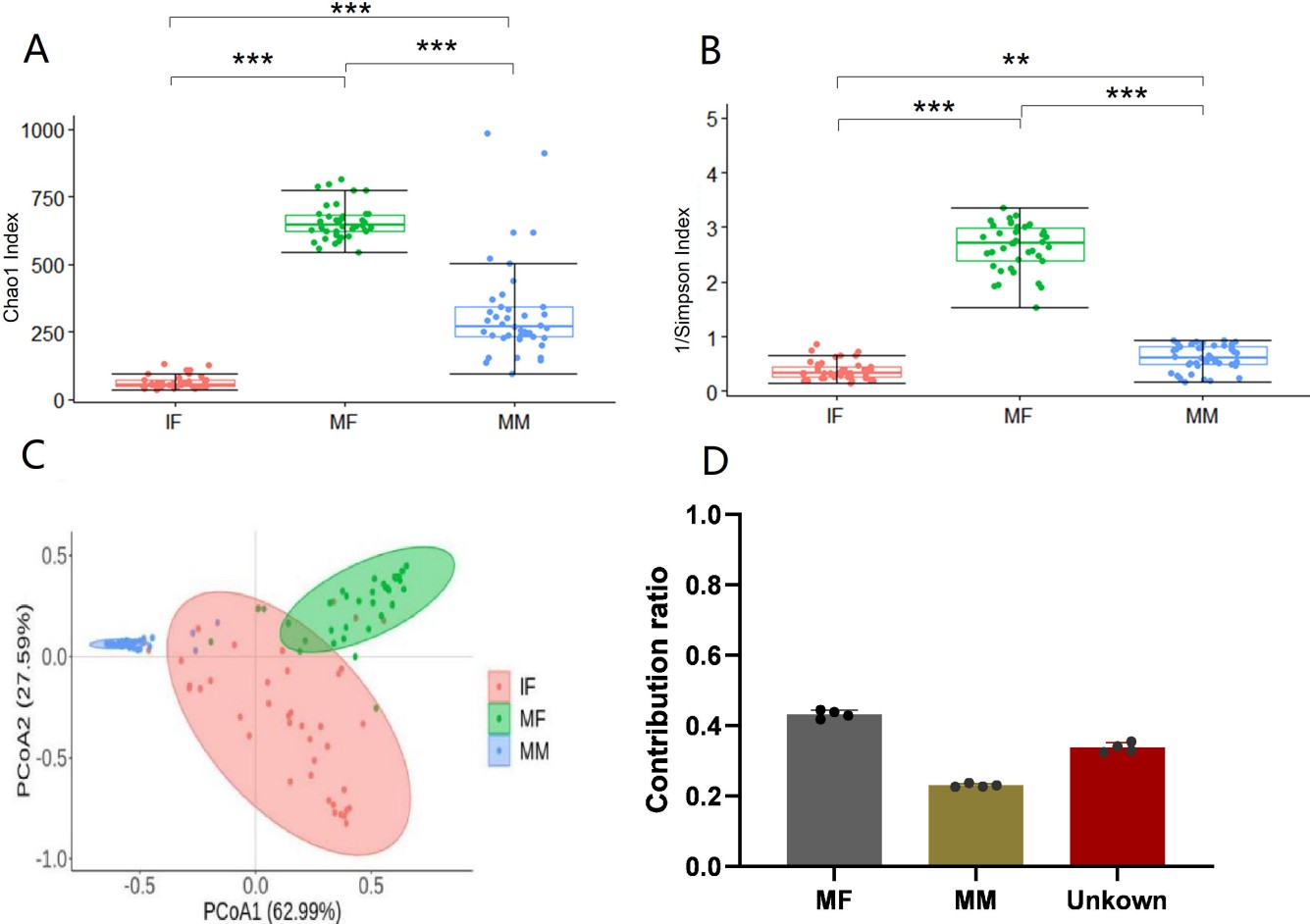

**FIG 3** Comparison of α- and β-diversity between the mother and infant samples and distribution of potential sources of gut microbiota in infants. (A and B) Significant differences in Chao1 and Inverse Simpson indices between maternal and infant samples (*P* < 0.001). (C) PCoA was performed between mother and infant samples according to the Bray–Curtis distance, and the three samples had an obvious separation effect. (D) The contribution rate of maternal intestinal microorganisms and breast milk microorganisms to infant intestinal microflora.

higher representation of maternal gut-associated bacterial taxa compared to those born via cesarean section. Upon utilizing maternal gut microbiota as the reference source for inferring the compositional makeup of the infant's gut microbiota, a statistically significant difference (*P* < 0.05) emerged, underscoring the differential impact of delivery mode on the initial colonization and assembly of the infant gut microbiome. Infants born via vaginal delivery were able to acquire more microbes and different microorganisms during labor than infants born via cesarean section. The gut microbial composition of mixed-fed infants appeared to be more adult-like and mature than that of breastfed children, which may be explained by the increased proportion of maternal gut microorganisms that contribute to the gut microbes of mixed-fed infants. Upon considering breast milk microbiota as a source, the extent of its incorporation into the gut microbiota of infants delivered vaginally or via cesarean section appeared comparably modest. Nevertheless, a striking observation emerged when contrasting the influence of breast milk microbes on exclusively breastfed infants vs those receiving mixed feeding: there was a statistically significant amplification (*P* < 0.0001) of the gut microbiota in infants who were solely nourished by breast milk, thereby highlighting the potent modulatory role of breast milk microbiota contingent upon the feeding regimen.

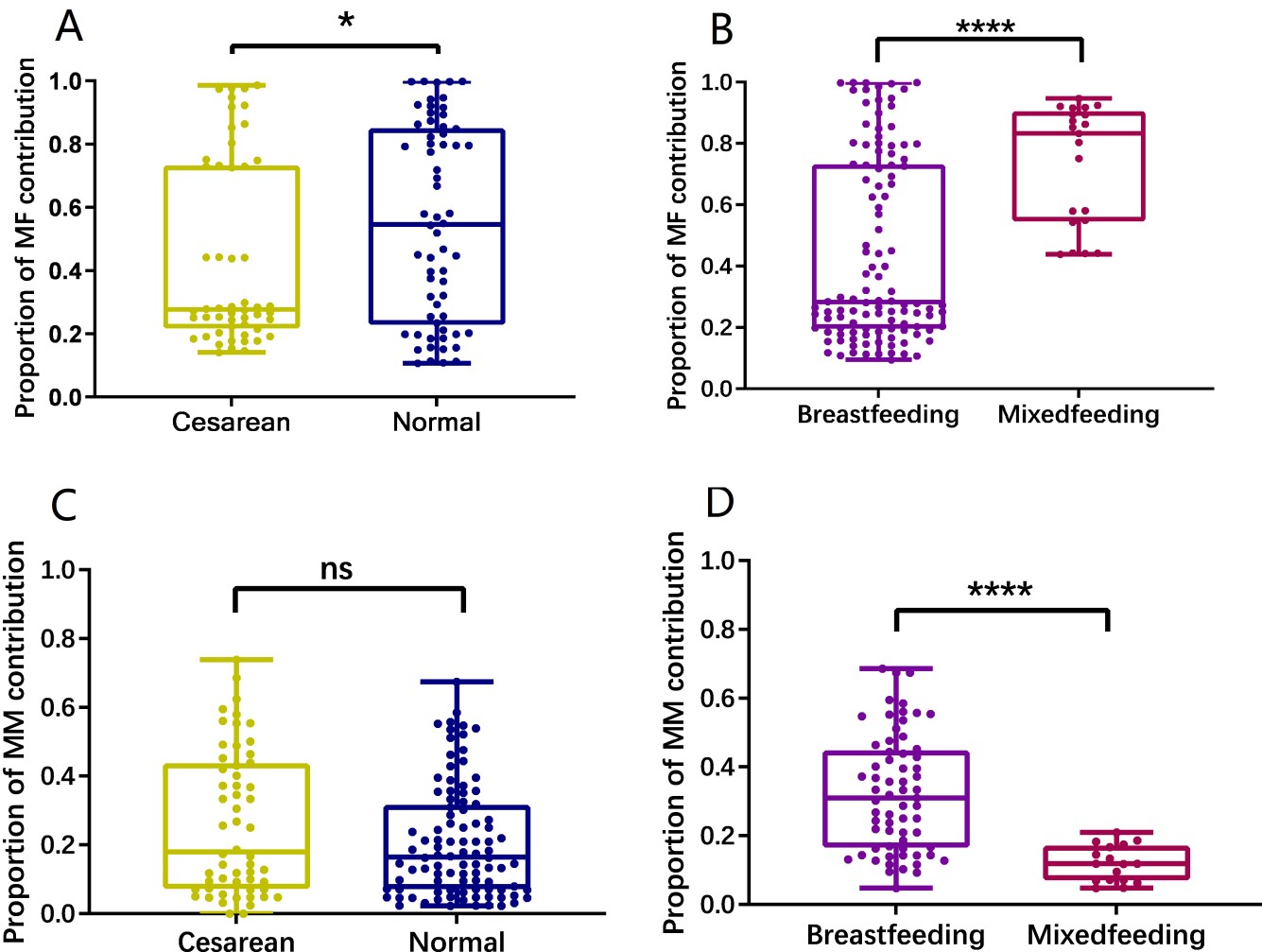

**FIG 4** Box plot of the contribution of the mother's samples from different sources to infant gut microbiota. (A) The significant difference in the contribution ratio of infant gut microbiota to different delivery methods using the maternal gut as the source ($P < 0.05$). (B) Very significant difference in the contribution ratio of infant gut microbiota to different feeding methods using the maternal gut as the source ($P < 0.0001$). (C) No significant difference in the contribution of breast milk microbiota as a source to the gut microbiota of infants with different delivery methods ($P = 0.1224$). (D) The significant difference in the contribution of breast milk microbiota as a source to the gut microbiota of infants with different feeding methods ($P < 0.0001$).

## DISCUSSION

In this study, we used the 16S rRNA gene sequencing method to investigate the early composition of gut microorganisms and microbial transfer between mother and infant pairs in infants aged 0–6 months of age. An examination of the origins of early-life gut microbiota, encompassing maternal gut microorganisms and those derived from breast milk, revealed that the former constitutes the predominant contributor to the infant's initial gut microbial composition, succeeded by breast milk-derived microorganisms and unascertained sources. This finding implies a significant transfer of maternal microbial constituents to the neonate's gastrointestinal tract. As demonstrated in earlier studies (7, 33–36), different compositional patterns at the phylum and genus levels were observed in all three samples in the association analysis of microbial communities in infant feces, maternal feces, and breast milk. Compared with previous reports (37), the present study did not find an increase in the α-diversity of gut microorganisms with infant growth during the early stages of life. The observed outcome could potentially stem from the limitations imposed by the modest sample size employed in this study or, alternatively,

the absence of a rigorous protocol for continuous sampling of identical specimens throughout the experimental duration.

Facultative anaerobic organisms, such as *Escherichia*, *Klebsiella*, and *Streptococcus*, initially colonize the microbiota during the first 6 months of life. These microorganisms are gradually replaced by specialized anaerobic organisms, such as *Bifidobacterium*, *Bacteroides*, and *Veillonella*, during the subsequent growth process. From 0 to 6 months of age, *Bifidobacterium* is predominant in the gut microbiota of infants, independent of whether they were full term at birth (37, 38), and infants exhibiting early *Bifidobacterium* presence in the gut have a lower risk of developing autoimmune disorders and intestinal inflammation (38, 39). While *Bifidobacterium* is known to be a prevalent inhabitant of the infantile gut during the early stages of life, a discernible upward trend in the *Klebsiella/Bifidobacterium* ratio among cesarean-born infants has been observed, a phenomenon that appears to bear a correlative relationship with the subsequent manifestation of allergic conditions during later childhood years (40). This may be an important link between the mode of delivery and the health of infants in later stages of growth. *Klebsiella* is a potentially harmful bacterium found in hospital environments. The study population likely obtained *Klebsiella* during surgery or their hospital stay. Individuals undergoing their formative years in highly unpredictable environments are prone to colonization by environmental microorganisms. The term "enterotype" was first used in 2011 and refers to three separate genera: *Bacteroides* (enterotype 1), *Prevotella* (enterotype 2), and *Ruminococcus* (enterotype 3) (41). However, the current study revealed that enterotype categories in infancy do not closely resemble adult enterotype categories. The microbiota *Bifidobacterium*, Proteobacteria, and Firmicutes dominated the intestinal phenotype of infants from 0 to 6 months of age. This characteristic differs from that in adults because *Bifidobacterium* is a member of the phylum Actinobacteria, and the subsequent bacterial evolution in infants may converge with that of adults. A study that gathered information on the intestinal microbiota of newborns during their first 3 years of life identified four significantly different gut types, with Firmicutes, *Bifidobacterium*, *Bacteroides*, and *Prevotella* dominating each. The four distinct gut configurations exhibit varying degrees of maturation, each aligning with specific phases of an individual's life course progression. Infants with a gut microbiota structure dominated by *Bacteroides* and *Prevotella* have a relatively stable gut microbiota structure, higher species diversity, and maturity. Infants with a gut microbiota structure dominated by Firmicutes and *Bifidobacterium* had a less stable gut microbial ecological structure and lower maturity. This was consistent with the later stages of newborn development. Xiao et al. noted that infants from diverse socioeconomic backgrounds have distinct dominant gut microbiota types (29). However, as the samples for this study were gathered primarily from Shijiazhuang, Hebei, no comparable conclusions could be drawn.

During the first 6 months, infants only consume breast milk directly from their mothers. Breast milk harbors microorganisms capable of infiltrating an infant's intestinal tract and is replete with immunoglobulins. The principal constituent is sIgA, a breast milk-specific immune protein and the inaugural immunological entity acquired by neonates. The immune systems of infants and young children grow gradually, and at 4–6 months after birth, the newborn's immune system, as well as other defenses against harmful microorganisms, are mostly obtained from sIgA in breast milk (42). Breastfed newborns can consume 0.5–1 g of sIgA per day, with concentrations in colostrum reaching as much as 15 mg/mL and in mature milk reaching just 1 mg/mL (43). Infant gut microbes were correlated with sIgA using Spearman's correlation coefficient because they may play a significant role in regulating and maintaining the development of immunity in infants. More recently, however, sIgA-coated bacteria have been studied, and their presence may signal disease (44). Actinobacteria, Bacteroidetes, Firmicutes, and Proteobacteria are the four major phyla present in infant intestinal microorganisms and have been identified in sIgA-coated microbiota (19). In this study, the dominant bacterial genera that were significantly correlated with sIgA belonged to these four phyla. In contrast, plasma cells triggered *in vivo* by gut microorganisms create sIgA in

the baby gut, which is then transported through the intestinal epithelium by the cellular polymeric immunoglobulin receptor (pIgR).

The establishment of the infant's gut microbiota plays an essential role during this early stage of life and is of tremendous importance for physical development later in life (45, 46). The close bond between mother and child is irreplaceable in nature. Infants are exposed to germs while they are still in the mother's womb and obtain them from maternal sources after delivery and nursing (47–49). The current study examined the contribution of two maternal sources, maternal gut microbes and breast milk microbes, to the infant's gut microbiota and found that the infant's microbiota was more closely linked to the maternal flora, indicating that both the mother's health during pregnancy and her physical state during infant feeding can affect infant growth. The results showed that the intestinal or breast milk microorganisms of the mother represent the main sources of infant intestinal microbiota. However, the mechanism underlying the acquisition of infant intestinal flora acquisition is not clear, and further research is still needed.

However, the contribution of maternal microorganisms tends to decrease as infants grow. Different modes of delivery and feeding can alter the composition of infant intestinal flora, thereby affecting the contribution of maternal microbes. The results of a cross-feeding experiment in mice showed that the gut microbiota of young mice was more similar to that of lactating mothers than to producing mothers, and the microbial composition formed in this case remained stable (50). In another study, the maternal gut microbiota was transplanted into cesarean section infants using the fecal microbiota transplantation (FMT) method. The results showed that the microbes of the cesarean section infants grew selectively after receiving maternal gut microbiota inoculation and were closer to the gut microbiota structure of the naturally delivered infants (51). The results of our investigation suggest that the intricate network architecture of the microbial community established during the first 6 months of life is fundamentally influenced by the intensity of intimate contact with the mother. Newborns delivered vaginally demonstrate a heightened acquisition of microorganisms originating from the maternal intestine, surpassing those acquired by infants delivered via cesarean section. Moreover, the gut microbial constitution of infants subjected to mixed feeding tends to align more closely with the maternal intestinal microbial profile, thus evidencing a convergent pattern.

This study had certain limitations. Owing to the extremely unstable structure and composition of the gut microbes in the early stages of life, our sample size may have been too small to accurately avoid small differences between individuals. Moreover, microorganisms in the mother's vagina, infant's mouth, and other parts were not performed, and they may represent the unknown source of infant intestinal microorganisms. Additionally, the employed 16S rRNA gene sequencing methodology, while informative, inherently possesses limitations in detecting low-abundance microbial taxa, thereby introducing potential biases into the outcomes. FEAST can efficiently quantify the fraction of each source environment (source) in the target microbial community (sink) (52); however, it is limited in its capacity to precisely elucidate the provenance and evolutionary trajectories of specific microbial strains, serving primarily as a predictive tool rather than a means of direct determination.

In summary, the gut microbiota of infants within the first 6 months of life is characterized by the presence of 11 distinct core microbial consortia, with Firmicutes (E1), *Bifidobacterium* (E2), and Proteobacteria exerting dominant influence over the overall microbial landscape. Notably, acetic acid, a key SCFA, exhibits the highest concentration within the infant's gut and displays a positive correlation with the abundance of *Megasphaera*. Furthermore, the levels of sIgA in the infant intestine show a positive association with *Bacteroides* populations, while exhibiting inverse correlations with *Klebsiella* and *Clostridium_XVIII*. Significant disparities in microbial richness and diversity are observed between maternal and infant samples, yet intriguingly, the gut microbiota of infants bears greater resemblance to that of their mother's intestinal microbiota. These

findings underscore the importance of maternal microbial transmission to the neonatal gut, a process with implications not only for the formulation of infant nutrition but also for the direct health outcomes of both the mother and her offspring.

## MATERIALS AND METHODS

### Study participants

Healthy mothers and their full-term infants were recruited between March and July 2022 in Shijiazhuang, Hebei Province, China. The recruitment criteria were as follows: (i) age <6 months; gestational age 37–42 weeks; singleton gestation; (ii) breastfeeding, mixed feeding, or artificial feeding; (iii) no antibiotic or probiotic preparation use for 2 weeks before sampling; (iv) mothers, free of dangerous diseases, had not undergone major gastrointestinal surgery within 5 years and had no abnormal eating habits; (v) mothers had not taken antibiotics or hormonal drugs within 3 months before sample collection. The exclusion criteria were as follows: (i) congenital malformations and congenital genetic metabolic diseases; (ii) feeding intolerance, diarrhea, gastrointestinal perforation, and gastrointestinal malformations. This study was approved by the Ethics Committee of Hebei Medical University (approval number: 20211111). All participants signed an informed consent form and received financial compensation.

### Sample collection

Fecal samples were collected from infants by collecting fresh feces from diapers after natural defecation using sterile disposable fecal cassettes. No toilets were used in the collection of the mothers' feces to prevent contamination of the samples. Breast pumps or hand milking was performed to collect the breast milk (sterilization was required for the tools used). These samples were delivered to the laboratory within 2 hours at 4°C for tube splitting and then frozen at −80°C. Detailed information on the infants enrolled in this study is presented in Table 4.

### DNA extraction and sequencing

The genomic DNA of each sample was extracted, and DNA quality was detected on 1% agarose gel. Variable regions V3–V4 of bacterial 16S rRNA gene were amplified with degenerate PCR primers, F338 (5′-ACTCCTACGGAGGGGGGGCAG-3′) and R806 (5′-GGAC-TACHVGGGTWTCTAAT-3′). PCR was performed by taking 30 ng of DNA samples of acceptable quality and the corresponding primers. The PCR amplification products were purified using Agencourt AMPure XP beads and dissolved in Elution Buffer. The fragment ranges and concentrations of the libraries were measured using an Agilent 2100 Bioanalyzer (Agilent, United States), and qualified libraries were sequenced. To create ASV sequences, sequences were deduplicated using QIIME2 software (version 2019.4) (53) and clustered at 100% similarity using the DADA2 method. The resulting signature sequences were collectively referred to as ASV. We compared particular ASV to the RDP (54) reference database with 99% sequence similarity. The microbiome survey in this study has adequate sequencing depth to depict the genuine image of the microorganisms in the samples, as evidenced by the mean coverage of the samples sequenced in this investigation, which was 99.99% ± 0.0017%. Q20 was used to evaluate the sequencing error rate. Sequencing results indicated that the number of bases reaching Q20 accounted for 98.4% (>90%).

### Determination of SCFAs and sIgA concentration

A handheld PHS-25 pH meter (INESA Science Instrument Co., Ltd., China) was used to measure the pH of the infant feces. A quantitative assessment of SCFAs in fecal samples using an external standard calibration method for gas chromatography (GC) has been reported (55). Briefly, at least, 0.2 g of feces was placed in a centrifuge tube, water

**TABLE 4** Detailed information on the study's infants

| Infant | Age (mo) | Ht (cm) | Wt (kg) | Gender | Mode of delivery | Feeding mode |
|---|---|---|---|---|---|---|
| 1 | 5 | 66 | 8 | Female | Cesarean | Exclusive breastfeeding |
| 2 | 5 | 68 | 9 | Male | Vaginal | Exclusive breastfeeding |
| 3 | 5 | 66 | 7.5 | Male | Vaginal | Mixed feeding |
| 4 | 5 | 73 | 9 | Male | Vaginal | Mixed feeding |
| 5 | 5 | 70 | 9 | Female | Cesarean | Exclusive breastfeeding |
| 6 | 3 | 60 | 6 | Male | Vaginal | Exclusive breastfeeding |
| 7 | 2 | 58 | 5.1 | Male | Vaginal | Mixed feeding |
| 8 | 3 | 68 | 6.8 | Female | Vaginal | Exclusive breastfeeding |
| 9 | 4 | 63 | 8 | Female | Vaginal | Exclusive breastfeeding |
| 10 | 4 | 69 | 8.8 | Female | Vaginal | Exclusive breastfeeding |
| 11 | 2 | 62 | 6 | Female | Cesarean | Exclusive breastfeeding |
| 12 | 2 | 62 | 6.5 | Female | Vaginal | Exclusive breastfeeding |
| 13 | 2 | 61 | 6 | Male | Cesarean | Exclusive breastfeeding |
| 14 | 6 | 73 | 9.2 | Female | Cesarean | Mixed feeding |
| 15 | 2 | 64 | 8 | Male | Vaginal | Exclusive breastfeeding |
| 16 | 3 | 63 | 6.5 | Female | Vaginal | Exclusive breastfeeding |
| 17 | 2 | 58 | 6 | Female | Vaginal | Exclusive breastfeeding |
| 18 | 3 | 63 | 6.5 | Male | Cesarean | Exclusive breastfeeding |
| 19 | 3 | 64 | 7.5 | Male | Cesarean | Exclusive breastfeeding |
| 20 | 1 | 60 | 5.4 | Female | Cesarean | Exclusive breastfeeding |
| 21 | 4 | 63 | 6 | Female | Cesarean | Mixed feeding |
| 22 | 4 | 61 | 7.5 | Female | Vaginal | Exclusive breastfeeding |
| 23 | 4 | 68 | 7.5 | Male | Cesarean | Exclusive breastfeeding |
| 24 | 3 | 62 | 7.1 | Male | Vaginal | Exclusive breastfeeding |
| 25 | 5.5 | 66 | 7.5 | Male | Vaginal | Exclusive breastfeeding |
| 26 | 4 | 64 | 6.6 | Male | Vaginal | Exclusive breastfeeding |
| 27 | 3 | 67 | 8.5 | Male | Vaginal | Exclusive breastfeeding |
| 28 | 5 | 64 | 9.5 | Male | Cesarean | Exclusive breastfeeding |
| 29 | 3.5 | 66 | 7 | Male | Vaginal | Exclusive breastfeeding |
| 30 | 4 | 65 | 7 | Female | Vaginal | Exclusive breastfeeding |
| 31 | 3.5 | 60 | 7.5 | Female | Cesarean | Exclusive breastfeeding |
| 32 | 5 | 70 | 7.5 | Female | Cesarean | Exclusive breastfeeding |
| 33 | 3 | 65 | 8 | Female | Vaginal | Exclusive breastfeeding |
| 34 | 3 | 60 | 7 | Female | Cesarean | Exclusive breastfeeding |
| 35 | 3 | 62 | 6.5 | Female | Vaginal | Exclusive breastfeeding |
| 36 | 4 | 64 | 7.5 | Female | Vaginal | Exclusive breastfeeding |
| 37 | 5 | 62 | 8.5 | Male | Vaginal | Exclusive breastfeeding |
| 38 | 4 | 62 | 6.5 | Female | Vaginal | Exclusive breastfeeding |
| 39 | 4 | 62 | 6.5 | Male | Vaginal | Exclusive breastfeeding |
| 40 | 1 | 62 | 6.1 | Male | Vaginal | Exclusive breastfeeding |
| 41 | 3 | 63 | 6.5 | Male | Vaginal | Mixed feeding |

was added to a volume of 1 mL, and then homogenization was performed. The pH was adjusted to 2.6–2.8 with 10 mM hydrochloric acid. The processed samples were centrifuged at 12,000 rpm (Eppendorf, Hamburg, Germany) for 15 min. The supernatant was collected and filtered through a sampling bottle using a water system filter membrane (Tianjin Jinteng Experimental Equipment Co., Ltd., PES, China, 0.22 µm). The supernatant (0.5 µL) was evaluated using a gas chromatography system (GC-FID, Agilent, USA) equipped with a DB-FFAP (Agilent, USA) column based on the following conditions: column temperature was ramped up to 180°C at 20°C/min for 1 min, then ramped up to 220°C at 50°C/min for 1 min; the split ratio was 10:1; and flow rate was 2.8 mL/min (nitrogen 99.99%). An IDK-sIgA ELISA kit (Immunodiagnostic, Germany) was used to

determine the content of secretory immunoglobulin A in infant feces (the procedure was strictly in accordance with the kit instructions).

## Statistical analysis

Data were analyzed using SPSS Statistics software (version 17.0). The $t$ test (normal distribution) and Mann–Whitney test (non-normal distribution) were used. In addition, the Wilcoxon test (two groups) or Kruskal-Wallis test (more than two groups) was used to analyze differences in the microbial diversity in different groups, and $P < 0.05$ was considered significant. The intestinal microbiota of newborns between the ages of 0 and 6 months was analyzed via principal component analysis. The primary composition of the human gastrointestinal microbiota is outlined by intestinal type (41). The Chao1 and Inverse Simpson indices were used to illustrate the α-diversity differences between mother and infant, and differences in microbial composition between maternal and neonatal samples (β-diversity) were described using PCoA based on Bray–Curtis distances. The main microbiota (specific OTUs) in maternal and infant samples was visualized using the parameter-specific occupancy rate (SPEC-OCCU), and microbiota (genus level) with specificity and occupancy >0.7 was determined. The R packages ggpubr, reshape2, and ggplot2 were used to calculate and display the indicators. The origin of the infant gut microbiota was examined using FEAST (52) software to predict and investigate whether the infant gut microbiota included the associated maternal microbiota. The 16S rRNA gene sequencing data type can be used in FEAST to effectively estimate thousands of contributions to the samples. This software compares sequences from newborn feces (sink samples) with those from breast milk and maternal feces (source samples) as the input samples. To assess the association between the dominant genera, SCFAs, and sIgA in the gut, calculations were performed using Spearman's correlation coefficient and visualized using heat maps.

## ACKNOWLEDGMENTS

We are grateful for the collaboration between the staff of the Technology Research Institute of the Junlebao Dairy Group and the study participants.

This study was funded by Shijiazhuang High Level Technology Innovation and Entrepreneurship Talent Project (grant number 07202202), Post-Subsidy Reward Special Fund Project in Shijiazhuang (grant number 226790217H), Hebei Province Key Cooperation Units with Academicians and Academician Workstation Projects (grant number 215A9919D), and Shijiazhuang Municipal Institute of Industrial Technology Construction Project (grant number 228790859A).

Conceptualization, M.L. and S.W.; Methodology, S.W., Y.X., and H.L.; Software, M.L.; Validation, J.B. and X.J.; Investigation, H.L. and L.C.; Data curation, M.L.; Writing—original draft, M.L.; Supervision, Q.Y. and Y.N.; Funding acquisition, S.W.

## AUTHOR AFFILIATIONS

[1]College of Food Science and Biology, Hebei University of Science and Technology, Shijiazhuang, Hebei, China
[2]Junlebao Dairy Group Co., Ltd., Shijiazhuang, Hebei, China
[3]Institute of Basic Medicine, Hebei Medical University, Shijiazhuang, Hebei, China

## AUTHOR ORCIDs

Shijie Wang 🔵 http://orcid.org/0000-0003-2856-5057

## FUNDING

| Funder | Grant(s) | Author(s) |
| --- | --- | --- |
| The Key R&D Program of Heibei | 21327110D | Shijie Wang |

| Funder | Grant(s) | Author(s) |
|---|---|---|
| Science and Technology Research and Development Program of Shijiazhuang | 216170107A | Shijie Wang |
| Shijiazhuang High Level Technology Innovation and Entrepreneurship Talent Project | 07202202 | Qingbin Yuan |

## DATA AVAILABILITY

The original contributions presented in the study are publicly available. These data can be found at BioProject PRJNA985881.

## ADDITIONAL FILES

The following material is available online.

### Open Peer Review

**PEER REVIEW HISTORY (review-history.pdf).** An accounting of the reviewer comments and feedback.

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
