## [Reviewer comments · Microbiology Spectrum]

Microbiology Spectrum

Relationship Between Infant Gastrointestinal Microorganisms and Maternal Microbiome Within 6 Months of Delivery

Li Menglu, Xue Yuling, Lu Han, Bai Jinping, Cui Liru, Ning Yibing, Yuan Qingbin, Jia Xianxian, and Wang Shijie

Corresponding Author(s): Wang Shijie, Hebei University of Science and Technology

Review Timeline:

Submission Date:	October 17, 2023
Editorial Decision:	November 9, 2023
Revision Received:	November 20, 2023
Editorial Decision:	February 14, 2024
Revision Received:	March 12, 2024
Editorial Decision:	March 13, 2024
Revision Received:	April 2, 2024
Accepted:	April 8, 2024

Editor: Erik Hom

Reviewer(s): Disclosure of reviewer identity is with reference to reviewer comments included in decision letter(s). The following individuals involved in review of your submission have agreed to reveal their identity: Jay Vornhagen (Reviewer #1)

Transaction Report:

DOI: <https://doi.org/10.1128/spectrum.03608-23>

Re: Spectrum03608-23 (Relationship Between Infant Gastrointestinal Microorganisms and Maternal Microbiome Within 6 Months of Delivery)

Dear Dr. Wang Shijie:

Thank you for the privilege of reviewing your work. Below you will find my comments, instructions from the Spectrum editorial office, and the reviewer comments.

I have reviewed your responses to the reviewers comments and you have not addressed the major concerns raised by Reviewer #2 in your prior mSphere submission. While we do not make decisions on novelty as is evaluated at mSphere, we do make judgments on scientific soundness and I feel the issues raised by Reviewer #2 need to be addressed in your revisions/rebuttal. Can you please do that, and also make sure how/where you revised your manuscript to address the concerns the reviewers raised? Only then will I be able to act on your manuscript.

Revision Guidelines

Sincerely,
Erik Hom
Editor
Microbiology Spectrum

Reviewer comments:

Reviewer #1 (Comments for the Author):

General comments:

Q:- Line 23-24:'this study showed that the microorganisms dominating the gut microbiota of 0-6 month infants were different from those reported previously'. How differently?

A: Thank you for this valuable comment. The text has been changed as follows: The results of this study showed that in addition to Firmicutes (E1) and Bifidobacterium (E2), the dominant microorganisms of the intestinal microbiota of infants aged 0-6 months include Proteobacteria, which is different from previous findings.

Q:- Line 51-52:'Another hypothesis holds that an infant's gut microbiota originates from the environment.' Please provide reference.

A: Thank you for this insightful comment. The expression was incorrect and has been modified as follows: Studies have also suggested that the environment and the mother's birth canal and excretions are significant sources of microbes for infants.

Q:- Line 44 and others:'gut flora' is an obsolete term, suggestion to replace as'gut microbiota'.

A: Thank you for this valuable comment. We have revised the text.

Q:- Line 64-65:'two parameters' instead of 'two materials'? Instead of recent years, SCFAs composition has been widely studied for years in gut microbiota research.

A: Thank you for this valuable comment. We have revised the text.

Q:- Line 67-69: Overall adult SCFAs composition is described here. The composition in infant feces is known to be different. Suggested reading: Pham et al. 2016, Early colonization of functional groups of microbes in the infant gut. Environ Microbiol.

A: Thank you for this valuable comment. The text has been changed as follows: The major SCFAs in the adult body are acetic acid (AA), propionic acid (PA), and butyric acid (BA), which account for more than 90% of the total acids and have a ratio of 3:1:1. However, the SCFA content in the intestines of infants differs from that in the intestine of adults and presents large individual differences (1).

Q:- Samples were available for 41 infants and 38 mothers. Please explain the missing maternal fecal samples and the impact on subsequent FEAST analysis.

A: Three of the forty-one infant fecal samples were collected twice at different time points, while the maternal fecal samples were only collected once. Because the relative abundance of each microorganism is considered to fluctuate over time in adults and vary between individuals, the microbial composition can remain stable for a long time (2). Moreover, the two maternal fecal samples were collected within a short time period. Ultimately, only infant feces and breast milk samples were collected more than once in the three mother-infant pairs.

Q:- Gut microbiota undergoes substantial changes within 0-6 months. As only single timepoint sampling was performed, please provide specific timing in which maternal and infant fecal samples were taken.

A: Before collecting maternal and infant samples, the mothers (samplers) were

provided uniform training. Since the babies were not able to autonomously defecate, the sampling time was set to collect the infant's feces immediately after defecation, while that of breast milk and the mother's feces was set within 2 hours after collecting the infant's feces.

Q:- Line 352:'mothers free of dangerous diseases'? What are the example of dangerous diseases?

A: China has stipulated that 36 major diseases represent dangerous diseases, including malignant tumors, major organ transplantation or hematopoietic stem cell transplantation, paralysis, loss of language ability, systemic lupus erythematosus, insulin-dependent diabetes (type I diabetes), and severe rheumatoid arthritis.

Q:- Line 362:'No toilets were used in the collection of the mothers' feces to prevent contamination of the samples'. Please explain how the samples collected instead.

A: When collecting the mother's stool samples, to prevent urine, sewage, and other impurities from mixing into the stool, we required the mother to urinate before defecation and use a clean squat toilet. Paper was placed on the bottom of the commode, and a clean piece of plastic wrap was placed on top of the lined paper, thus leaving the feces on top of the plastic wrap. The feces in the middle part were selected and collected in a sterile tube.

Q:- Line 374-377: Amplicon analysis was performed using DADA2. These are ASVs instead of OTUs.

A: Thank you for highlighting this. We have revised the text.

Q:- Was sample collection performed during hospital visit or at home?

A: It was performed home.

Q:- Line 131-132: What does'the in vitro experimental methods' refer to?

A: The in vitro experimental method mentioned here refers to the follow-up experiment of this study. To explore the effect of breast milk oligosaccharides on infant intestinal flora, an in vitro fermentation method was adopted.

Q:- Line 132-133: Suggestion to use Amsterdam infant stool scale to access characteristics of infant stools.

A: The amount, consistency, and color of infant feces were described by using the Amsterdam infant stool scale (3). The infant feces were different in shape and primarily had values of Amount = 2, Consistency = B, Color = II.

Q:- There is no mentioning of lactic acid in infant feces. Is this not analyzed or not detected?

A: We did not test for lactic acid in the infant feces because we mainly focused on the relationship between short-chain fatty acids and secretory immunoglobulin A and gut microbiota.

Q:- The analysis on the contribution of maternal to infant gut microbiota was performed solely on V3-V4 16S rRNA sequencing result on fecal and breastmilk samples. There is limitation for 16S gene to provide taxonomic resolution at species and strain level. Please consider full-length 16S intragenomic copy variants.

A: Although we understand the shortcomings of 16S rRNA gene sequencing, we felt that it was an appropriate method based on the actual situation. In later experiments we will consider using full-length 16S intragenomic copy variants because they have

the potential to provide a taxonomic resolution of bacterial communities at the species and strain levels (4).

Q:- There are several grammatical errors, awkward sentence structures, and instances of unclear or ambiguous language. Please proofread the full manuscript.

A: We have proofread the entire manuscript to the best of our abilities and apologize for the inconvenience.

Specific comments:

Q:- Line 20: Acronyms for 'FEAST' should be spelt out in full the first time it is used.

A: Thank you for this valuable comment. We have revised the text.

Q:- Line 25: 'significantly positively'. Use of two adverbs consecutively. Significantly can be dropped as p value is provided at the end of sentence.

A: Thank you for this valuable comment. We have revised the text.

Q:- Line 35-36: 'infant gut microbes contributed via vaginal delivery and mixed feeding'? Please rephrase.

A: Thank you for this valuable comment. We have revised the text as follows: The gut microbiome of infants produced via vaginal delivery and mixed feeding is more similar to that of the maternal gut microbiota.

Q:- Line 40-42: 'play a significant physiological role in the whole body' and 'it is closely related to various physiological activities of the body'. Repeat of a similar message in one sentence. Please rephrase.

A: Thank you for this valuable comment. We have revised the text.

Q:- Line 75: 'many grams of sIgA'? How many?

A: The intestinal mucosa contains the largest population of antibody-secreting plasma cells in the body. However, to our knowledge, the amount of sIgA released by plasma cells has not been clearly reported. The content of sIgA in colostrum is as high as 15 mg/ml, and the concentration in mature milk is 1 mg/ml. Breastfed infants can consume 0.5-1 g/day (5).

Q:- Line 402: 'infant,,'

A: Thank you for this valuable comment. We have revised the text.

Q:- Caption of Figure 1 in tracked changes mode.

A: Thank you for this valuable comment. We have revised the text.

Q:- Table 3: Please revise the unit for sIgA.

A: Thank you for this valuable comment. We have revised the text.

Reviewer #2 (Comments for the Author):

Minor concerns

Q1. Line 139. Please define VA before using the acronym.

A: Thank you for this valuable comment. VA represents valeric acid, and the text has been revised.

Q2. Table 2. Please clarify what "Coverage (%)" means. Does this refer to the number

of samples where this OTU was detected?

A: 'Coverage (%)' represents the coverage of core species in the sample. For example, Coverage=100% for *Bifidobacterium* means that *Bifidobacterium* was present in all 41 infant feces; Coverage=97.56% for *Megasphaera* means that *Megasphaera* was present in 39 out of 41 samples.

3. Line 163. I am unclear what the authors mean by "sequenced species."

A: We apologize for any confusion caused by our misrepresentation. The revised statement is as follows: The microbial populations in breast milk (MM), maternal feces (MF), and baby feces (IF) differed (Figure 2), and the four major phyla that accounted for more than 90% of the relative abundance of the intestinal flora were illustrated in the figure (Figure 2 B).

Q:4. Line 191 and Fig. 3B. I believe this is the Inverse Simpson Index.

A: The index in the cited text is the Simpson Index rather than the Inverse Simpson Index. The Simpson index was used to represent the species diversity in the community.

Q:5. Line 224 and Figs. 4A, C. Please use the term "Vaginal" rather than "Normal" when referring to vaginal births.

A: Thank you for this valuable comment. We have revised the text.

Q:6. Lines 252-255. I am confused by this section. Where are these data presented? From my understanding this is a cross-sectional study, thus no temporal conclusions can be made.

A: In this section we ignored interindividual differences and grouped the population according to the criteria of age in months, with 0–6 months divided into three time stages: 0–2 months, 3–4 months, and 5–6 months.

Q:7. Line 366. Do the authors mean aliquoting instead of tube splitting?

A: Aliquoting was not performed. Rather, the samples were divided into three tubes for microbial diversity detection.

Q:8. Line 403. Typo in this sentence.

A: Thank you for this valuable comment. We have revised the text.

Q:9. Methods section "DNA Extracting and Sequencing." Did the authors assess their sequencing error rate? If so, please report that rate.

A: Yes, we have evaluated the sequencing error rate by Q20. The sequencing results showed that the number of bases reaching Q20 accounted for 98.4% (not lower than 90%).

Additions to Reviewer #2

Major Concerns

Q1. Throughout the manuscript, the authors imply the existence of a placental microbiome and that microbiome seeding occurs in utero. These concepts are, at best, controversial and are rejected by most researchers who focus on the maternal-fetal microbiome (see PMID 36934229 as an example). Given that the authors' focus is the fecal and breast milk microbiome, it is unnecessary to wade into this controversy. Rather, it is more appropriate that the authors focus on the data they present and limit their conclusions to those data.

A: Thank you for this insightful comment. I have removed the reference to the

placental microbiome as suggested. The conclusions have focused on the link between the mother and infant microbiomes.

Q2. There are several instances where the authors use language that lacks specificity or clarity. For example, in lines 30-32, the authors conclude that their data have implications for the "mother's diet and lifestyle." No data is presented to support any such conclusions, as the mother's diet was not explored in detail nor it is clear what lifestyle is referring to. Examples of other non-specific language include referring to the "gut microbiota" when the fecal microbiota is being interrogated, using the term "16S rRNA sequencing" instead of "16S rRNA gene sequencing," and use of the word "species" instead of "OTU". These are not the only instances, thus I encourage the authors to revise their manuscript to refer to the data that are presented.

A: Thank you for this insightful comment. Regarding mothers' diets and lifestyles during the collection of the questionnaire we summarized the analysis. The difference in the contribution of mothers with different BMI to the gut microbiota of their infants was not significant ($p=0.098$). I apologize for any presentation errors in the text. I have corrected the errors in the text.

Q3. The figures are referenced out of order and lack important elements for interpretation. Please specify the figure legends and symbols in Figs. 1B-C, modify Figs. 2B-C to make these figures more legible, and add individual data points to Figs. 3D and 4.

A: Thank you for this valuable comment. We have revised the figures. In this case, the four dots on each bar in Figure 3-D indicate that four parallels were made when exploring the specific contribution of each source to the infant gut microbiota.

Q4. Please use ANOVA and post hoc multiple comparison testing when comparing >2 groups in Figs. 3A-B.

A: Thank you for this valuable comment. Because the data did not fit a normal distribution, a multiple independent samples Kruskal-Wallis test was used for Figs. 3A-B. Post hoc multiple comparisons were also conducted.

1. Pham VT, Lacroix C, Braegger CP, Chassard C. 2016. Early colonization of functional groups of microbes in the infant gut. *Environmental Microbiology* 18:2246-2258.
2. Fassarella M, Blaak EE, Penders J, Nauta A, Zoetendal EG. 2020. Gut microbiome stability and resilience: elucidating the response to perturbations in order to modulate gut health. *Gut* 70:595-605.
3. Bekkali N, Hamers SL, Reitsma JB, Van Toledo L, Benninga MA. 2008. Infant Stool Form Scale Development and Results. *The Journal of Pediatrics* 154:521-526.
4. Johnson JS, Spakowicz DJ, Hong BY, Petersen LM, Weinstock GM. 2019. Evaluation of 16S rRNA gene sequencing for species and strain-level microbiome analysis. *Nature Communications* 10.
5. Dunne-Castagna Vanessa P, Mills David A, Lönnerdal B. 2020. Effects of Milk Secretory Immunoglobulin A on the Commensal Microbiota. *Nestle Nutrition Institute Workshop Series* 94:158-168.

Re: Spectrum03608-23R1 (Relationship Between Infant Gastrointestinal Microorganisms and Maternal Microbiome Within 6 Months of Delivery)

Dear Dr. Wang Shijie:

Thank you for the privilege of reviewing your work. Below you will find my comments, instructions from the Spectrum editorial office, and the reviewer comments.

Please take great care in addressing the reviewer's comments; I cannot make a favorable decision in your favor if not addressed! I will give you a chance to do this before I make a final decision.

Revision Guidelines

Sincerely,
Erik Hom
Editor
Microbiology Spectrum

Reviewer #1 (Comments for the Author):

Several of my comments have not been addressed.

1. Lines 48-49 still indicate the presence of a placental microbiome.

2. A significant amount of non-specific language is still present, including language I specifically referred to, for example, "normal" instead of "vaginal" in Figure 2, "16S rRNA sequencing" instead of "16S rRNA gene sequencing," and use of the word "species" instead of "OTU". There are more instances not listed in the examples above.

3. The conclusions regarding the data in Figure 3B are not supported if they are Simpson Index, as the authors purport in their response. The Simpson Index (D) = $[\sum(n(n-1))]/[N(N-1)]$, where n = number of reads of each ASV and N = total number of reads of all ASVs. Thus, a D closer to 0 would have higher diversity than that closer to 1, and always be a value between 0 and 1. I suggested that what is shown in Fig. 3B is $1/D$ (Inverse Simpson index); however, if the authors state that this is the Simpson Index, then there is a calculation error.

4. The sequencing error rate is not reported in the manuscript as was requested.

5. Regarding responses to reviewer 1, I do not feel that the following comments were adequately addressed:

"Gut microbiota undergoes substantial changes within 0-6 months. As only single timepoint sampling was performed, please provide specific timing in which maternal and infant fecal samples were taken." The authors did not indicate the infant age at which the fecal sample was collected.

"Samples were available for 41 infants and 38 mothers. Please explain the missing maternal fecal samples and the impact on subsequent FEAST analysis." Using sequential samples may lead to bias not discussed in the manuscript. Alternatively, the authors should consider limiting their analysis to unique samples.

Thank you very much for these valuable suggestion. According to the nice suggestions, we have made extensive corrections to our previous draft. And here we did not list the changes but marked in red in the revised paper. We appreciate for Reviewers' warm work earnestly and hope that the correction will meet with approval.

Reviewer #1 (Comments for the Author):

Comment-1. Lines 48-49 still indicate the presence of a placental microbiome.

Reply: Thanks for pointing out the problem, we have revised the text.

Comment-2. A significant amount of non-specific language is still present, including language I specifically referred to, for example, "normal" instead of "vaginal in Figure 2, "16S rRNA sequencing" instead of "16S rRNA gene sequencing," and use of the word "species" instead of "OTU". There are more instances not listed in the examples above.

Reply: Thank you for this valuable comment. We were really sorry for our careless mistakes. We have revised the text.

Comment-3. The conclusions regarding the data in Figure 3B are not supported if they are Simpson Index, as the authors purport in their response. The Simpson Index ($D = [\sum(n(n-1))]/[N(N-1)]$), where n = number of reads of each ASV and N = total number of reads of all ASVs. Thus, a D closer to 0 would have higher diversity than that closer to 1, and always be a value between 0 and 1. I suggested that what is shown in Fig. 3B is $1/D$ (Inverse Simpson index); however, if the authors state that this is the Simpson Index, then there is a calculation error.

Reply: We were really sorry for our careless mistakes. We are confident that the Figure 3B represents the Inverse Simpson index.

Comment-4. The sequencing error rate is not reported in the manuscript as was requested.

Reply: Thank you for this valuable comment. We have revised the text (Page 12 ,Line 371).

Editor:

Comment-5. Regarding responses to reviewer 1, I do not feel that the following comments were adequately addressed:

(1) -"Gut microbiota undergoes substantial changes within 0-6 months. As only single timepoint sampling was performed, please provide specific timing in which maternal and infant fecal samples were taken." The authors did not indicate the infant age at which the fecal sample was collected.

Reply: Thanks for pointing out the problem, we are sorry for our careless mistakes. The detailed information on infant fecal samples please see Table 4. The table has been included in the manuscript (Page 10 ,Line 355).

Table 4 Detailed information on infant fecal samples

Age(month)	Hight	Weight	Gender	Mode of delivery	Feeding mode
------------	-------	--------	--------	------------------	--------------

		(cm)	(kg)			
1	5	66	8	Female	Cesarean	Exclusive breastfeeding
2	5	68	9	Male	Normal	Exclusive breastfeeding
3	5	66	7.5	Male	Normal	Mixed feeding
4	5	73	9	Male	Normal	Mixed feeding
5	5	70	9	Female	Cesarean	Exclusive breastfeeding
6	3	60	6	Male	Normal	Exclusive breastfeeding
7	2	58	5.1	Male	Normal	Mixed feeding
8	3	68	6.8	Female	Normal	Exclusive breastfeeding
9	4	63	8	Female	Normal	Exclusive breastfeeding
10	4	69	8.8	Female	Normal	Exclusive breastfeeding
11	2	62	6	Female	Cesarean	Exclusive breastfeeding
12	2	62	6.5	Female	Normal	Exclusive breastfeeding
13	2	61	6	Male	Cesarean	Exclusive breastfeeding
14	6	73	9.2	Female	Cesarean	Mixed feeding
15	2	64	8	Male	Normal	Exclusive breastfeeding
16	3	63	6.5	Female	Normal	Exclusive breastfeeding
17	2	58	6	Female	Normal	Exclusive breastfeeding
18	3	63	6.5	Male	Cesarean	Exclusive breastfeeding
19	3	64	7.5	Male	Cesarean	Exclusive breastfeeding
20	1	60	5.4	Female	Cesarean	Exclusive breastfeeding
21	4	63	6	Female	Cesarean	Mixed feeding
22	4	61	7.5	Female	Normal	Exclusive breastfeeding
23	4	68	7.5	Male	Cesarean	Exclusive breastfeeding
24	3	62	7.1	Male	Normal	Exclusive breastfeeding
25	5.5	66	7.5	Male	Normal	Exclusive breastfeeding
26	4	64	6.6	Male	Normal	Exclusive breastfeeding
27	3	67	8.5	Male	Normal	Exclusive breastfeeding
28	5	64	9.5	Male	Cesarean	Exclusive breastfeeding
29	3.5	66	7	Male	Normal	Exclusive breastfeeding
30	4	65	7	Female	Normal	Exclusive breastfeeding
31	3.5	60	7.5	Female	Cesarean	Exclusive breastfeeding
32	5	70	7.5	Female	Cesarean	Exclusive breastfeeding
33	3	65	8	Female	Normal	Exclusive breastfeeding
34	3	60	7	Female	Cesarean	Exclusive breastfeeding
35	3	62	6.5	Female	Normal	Exclusive breastfeeding
36	4	64	7.5	Female	Normal	Exclusive breastfeeding
37	5	62	8.5	Male	Normal	Exclusive breastfeeding
38	4	62	6.5	Female	Normal	Exclusive breastfeeding
39	4	62	6.5	Male	Normal	Exclusive breastfeeding
40	1	62	6.1	Male	Normal	Exclusive breastfeeding
41	3	63	6.5	Male	Normal	Mixed feeding

(2) "Samples were available for 41 infants and 38 mothers. Please explain the

missing maternal fecal samples and the impact on subsequent FEAST analysis." Using sequential samples may lead to bias not discussed in the manuscript. Alternatively, the authors should consider limiting their analysis to unique samples.

Reply: To minimize such effects, duplicate infant samples (sample number: 39, 40, 41) were systematically excluded from our analysis. We proceeded with sourcing attribution on a meticulously curated dataset of microbiota from 38 mother-newborn dyads. The conclusive findings revealed that the maternal gut (at $43.24\% \pm 11.16\%$) served as the primary reservoir for the neonatal gut microbiota during early infancy, with a notably lesser contribution from breast milk ($23.01\% \pm 0.47\%$) (Figure 3D). Similarly, when assessing the influence of delivery mode and feeding method on the source, we have also made corresponding adjustments (Figure 4).

Figure 3D. The contribution rate of maternal intestinal microorganisms and breast milk microorganisms to infant intestinal microflora

Re: Spectrum03608-23R2 (Relationship Between Infant Gastrointestinal Microorganisms and Maternal Microbiome Within 6 Months of Delivery)

Dear Dr. Wang Shijie:

Thank you for the privilege of reviewing your work. Below you will find my comments, instructions from the Spectrum editorial office, and the reviewer comments.

I appreciate your revisions, but unfortunately, I believe you misunderstood the Reviewer's comments:

"Comment-2. A significant amount of non-specific language is still present, including language I specifically referred to, for example, "normal" instead of "vaginal in Figure 2, "16S rRNA sequencing" instead of "16S rRNA gene sequencing," and use of the word "species" instead of "OTU". There are more instances not listed in the examples above."

You were asked to replace "normal" with "vaginal, NOT the other way around! (You have now have "normal" for every instance of vaginal birth/delivery but the issue is that "normal" is not a proper scientific description of births.) Similarly, you were not supposed to use "species" but rather "OTU", but you still have "species" in the manuscript. It would be wise to review Reviewer #1's original review comments to help you with context and in carefully understanding their suggestions and criticisms.

Importantly, there are still grammatical/writing issues that prevent this manuscript from being acceptable. For example, in lines 316-317 you write: "Infants delivered normal delivery obtain more microorganisms from the maternal intestine..." -- "delivered normal delivery obtain more" is very poor English. Please do not merely "find and replace" the word replacements suggested by the Reviewer but make sure to double check the English language surrounding these problem spots make sense.

Your references also have some errors. E.g., see ref 4, 5, 32 (all caps), etc. If I can spot these errors with a cursory inspection, there are certainly other errors.

This will be the final revision I will permit; please be extra careful in your revisions; I suggest you carefully revise and/or have a language service (<https://journals.asm.org/writing-your-paper?journal=spectrum#language-editing-services>) or native speaker correct lingering errors. While the manuscript is improved from the original version you submitted, it could still be brought to a higher standard.

In your cover letter to me, please make sure you detail very clearly and specifically everything you have changed so as to facilitate the decision process.

Revision Guidelines

Data availability: ASM policy requires that data be available to the public upon online posting of the article, so please verify all links to sequence records, if present, and make sure that each number retrieves the full record of the data. If a new accession number is not linked or a link is broken, provide Spectrum production staff with the correct URL for the record. If the accession

numbers for new data are not publicly accessible before the expected online posting of the article, publication may be delayed; please contact production staff (Spectrum@asmusa.org) immediately with the expected release date.

Sincerely,
Erik Hom
Editor
Microbiology Spectrum

Reply to the editor's comments:

Thank you very much for your valuable suggestion. According to your nice suggestions, we have made extensive corrections to our previous manuscript. And here we did not list the changes but marked in red in the revised paper (including text and references). We appreciate for yours warm work earnestly and hope that the correction will meet with approval.

Re: Spectrum03608-23R3 (Relationship Between Infant Gastrointestinal Microorganisms and Maternal Microbiome Within 6 Months of Delivery)

Dear Dr. Wang Shijie:

Thank you for your edits and revisions. I have reviewed your revisions and am accepting your revised manuscript - I am forwarding it to the ASM production staff for publication. (In the future, I suggest carefully editing and addressing the reviewers' comments as meticulously as possible in the first pass as that would be to your advantage in the review process!)

Your paper will first be checked to make sure all elements meet the technical requirements. ASM staff will contact you if anything needs to be revised before copyediting and production can begin. Otherwise, you will be notified when your proofs are ready to be viewed.

Sincerely,
Erik Hom
Editor
Microbiology Spectrum